# COBE: Contextualized Object Embeddings from Narrated Instructional Video

**Gedas Bertasius**[1], **Lorenzo Torresani**[1,2]
[1]Facebook AI, [2]Dartmouth College

## Abstract

Many objects in the real world undergo dramatic variations in visual appearance. For example, a tomato may be red or green, sliced or chopped, fresh or fried, liquid or solid. Training a single detector to accurately recognize tomatoes in all these different states is challenging. On the other hand, contextual cues (e.g., the presence of a knife, a cutting board, a strainer or a pan) are often strongly indicative of how the object appears in the scene. Recognizing such contextual cues is useful not only to improve the accuracy of object detection or to determine the state of the object, but also to understand its functional properties and to infer ongoing or upcoming human-object interactions. A fully-supervised approach to recognizing object states and their contexts in the real-world is unfortunately marred by the long-tailed, open-ended distribution of the data, which would effectively require massive amounts of annotations to capture the appearance of objects in all their different forms. Instead of relying on manually-labeled data for this task, we propose a new framework for learning **C**ontextualized **OB**ject **E**mbeddings (COBE) from automatically-transcribed narrations of instructional videos. We leverage the semantic and compositional structure of language by training a visual detector to predict a contextualized word embedding of the object and its associated narration. This enables the learning of an object representation where concepts relate according to a semantic language metric. Our experiments show that our detector learns to predict a rich variety of contextual object information, and that it is highly effective in the settings of few-shot and zero-shot learning.

## 1 Introduction

In recent years, the field of object detection has witnessed dramatic progress in the domain of both images [1, 2, 3, 4] and videos [5, 6, 7, 8, 9]. To a large extent these advances have been driven by the introduction of increasingly bigger labeled datasets [10, 11, 12, 13], which have enabled the training of progressively more complex and deeper models. However, even the largest datasets in this field [10, 11, 12] still define objects at a very coarse level, with label spaces expressed in terms of nouns, e.g., tomato, fish, plant or flower. Such noun-centric ontologies fail to represent the dramatic variations in appearance caused by changes in object "states" for many of these classes. For example, a tomato may be fresh or fried, large or small, red or green. It may also frequently appear in conjunction with other objects such as a knife, a cutting board, or a cucumber. Furthermore, it may undergo drastic appearance changes caused by human actions (e.g., slicing, chopping, or squeezing).

The goal of this work is to design a model that can recognize a rich variety of contextual object cues and states in the visual world (see Figure 1 for a few illustrative examples). A detection system capable to do so is useful not only for understanding the object functional properties, but also for inferring present or future human-object interactions. Unfortunately, due to the long-tailed, open-ended distribution of the data, implementing a fully-supervised approach to this problem is challenging as it requires collecting large amounts of annotations capturing the visual appearance of objects in all their different forms. Instead of relying on manually-labeled data, we propose a

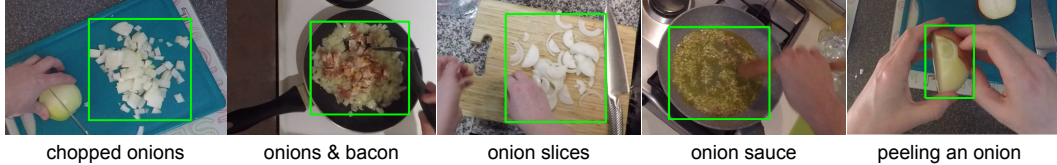

| chopped onions | onions & bacon | onion slices | onion sauce | peeling an onion |

Figure 1: Many objects in the real-world undergo state changes that dramatically alter their appearance, such as in the case of the onions illustrated in this figure. In this work, we aim at training a detector that recognizes the appearances, states and contexts of objects using narrated instructional video.

novel paradigm that learns **C**ontextualized **OB**ject **E**mbeddings (COBE) from instructional videos with automatically-transcribed narrations [14]. The underlying assumption is that, due to the tutorial properties of instructional video, the accompanying narration often provides a rich and detailed description of an object in the scene in terms of not only its name (noun) but also its appearance (adjective or verb), and contextual objects (other nouns) that tend to occur with it. We refer collectively to these ancillary narrated properties as the "contextualized object embedding" in reference to the contextualized word representation [15] which we use to automatically annotate objects in a video.

Specifically, we propose to train a visual detector to map each object instance appearing in a video frame into its contextualized word representation obtained from the contextual narration. Because the word representation is contextualized, the word embedding associated to the object will capture not only the category of the object but also the words that were used to describe it, e.g., its size, color, and function, other objects in the vicinity, and/or actions being applied to it. Consider for example the frame illustrated in Figure 2. Its accompanying narration is "... break an egg into a bowl..." The contextualized word embedding of the egg represented in the frame will be a vector that encodes primarily the word "egg" but also the contextual action of "breaking" and the contextual object "bowl." By training our model to predict the contextualized embedding of an object, we force it to recognize fine-grained contextual information beyond the coarse categorical labels. In addition to the ability of representing object concepts at a finer grain, such a representation is also advantageous because it leverages the compositionality and the semantic structure of the language, which enable generalization of states and contexts across categories.

We train COBE on the instructional videos of HowTo100M dataset [14], and then test it on the evaluation sets of HowTo100M, EPIC-Kitchens [16], and YouCook2 [17] datasets. Our experiments show that on all three of these datasets, COBE outperforms the state-of-the-art Faster R-CNN detector. Furthermore, we demonstrate that despite a substantial semantic and appearance gap between HowTo100M and EPIC-Kitchens datasets, COBE successfully generalizes to this setting where it outperforms several highly competitive baselines trained on large-scale manually-labeled data. Lastly, our additional experiments in the context of zero-shot and few-shot learning indicate that COBE can generalize concepts even to novel classes, not seen during training, or from few examples.

## 2   Related Work

**Object Detection in Images.**  Modern object detectors [1, 2, 3, 4, 18, 19, 20, 21, 22, 23, 24] are predominantly built on deep CNNs [25, 26, 27]. Most of these systems are typically trained on datasets where the label space is discretized, the number of object categories is fixed a priori, and where thousands of bounding box annotations are available for each object class. In contrast, our goal is to design a detection system that works effectively on datasets that exhibit long-tailed and open-ended distribution, and that generalizes well to the settings of few-shot and zero-shot learning. We accomplish this by designing a detection model that learns to map each visual object instance into a language embedding of that object and its contextual narration.

**Object States and Contexts.**  In the past, several methods have tackled the problem of object attribute classification [28, 29, 30, 31], which could be viewed as a form of object state recognition. However, unlike our approach, these methods assume a predefined set of attributes, and the existence of manually labeled datasets to learn to predict these attributes. Other methods [32, 33] aim to model object appearance changes during human-object interactions. However, due to their reliance on manual annotations, they are trained on small-scale datasets that contain few object categories and states. Finally, we note the approach in [34], which discovers object states and transformations from manually-annotated Web images. In comparison, our model does not require manual annotations of

object states and contexts, but instead relies on automatically-transcribed narrations from instructional videos. Furthermore, compared to the method in [34], our COBE operates in a continuous label space instead of in a discretized space. This allows COBE to be effective in few-shot and zero-shot learning. Lastly, our model is designed to produce an output for every detected object instance in a video frame, instead of producing a single output for the whole image [34].

**Vision, Language, and Speech.** Many prior methods for jointly modeling vision and language learn to project semantically similar visual and language inputs to the same embedding [14, 35, 36, 37, 38, 39, 40, 41, 42, 43]. These methods compute a single feature vector for the whole image or video, whereas our goal is to learn a representation for object instances detected in a given video frame. This allows us to capture fine-grained state, appearance and context cues relating to object instances. Closer to our goal, the method of Harwath et al. [44] leverages spoken audio captions to learn to localize the relevant portions of the image, and the work of Zhou et al. [17] uses textual video descriptions for visual object grounding. In contrast, our system: learns from transcribed speech of real-world Web videos rather than from verbal or textual descriptions provided ad-hoc by annotators; it goes beyond mere localization by predicting the semantics of contextual narration; and, finally, it leverages the semantic structure of a modern contextualized word model.

## 3 Technical Approach

Our goal is to design a system that predicts a contextualized object embedding for every detected object instance in a video frame. To train this model, we need ground-truth bounding boxes for all object instances appearing in frames of HowTo100M, our training dataset. Unfortunately such annotations are not available. Similarly to [45, 46], we address this issue by leveraging a pretrained model to transfer annotations from a labeled dataset to our unlabeled dataset. Afterwards, we train COBE using the obtained "pseudo" bounding box annotations, and the narrated speech accompanying each HowTo100M video frame. The next subsections provide the details of the approach.

### 3.1 Augmenting HowTo100M with Bounding Boxes

Inspired by recent semi-supervised learning methods [45, 46], we use a simple automatic framework to augment the original HowTo100M dataset with pseudo ground-truth bounding boxes using a pretrained detector. As our objective is to model object appearances in settings involving human-object interactions, we choose to leverage an object detector pretrained on a set of manually-labeled bounding boxes of EPIC-Kitchens [16] which is one the most extensive datasets in this area and is also a part of the benchmark we chose for our quantitative evaluations. The detector is trained to recognize the 295 object categories of EPIC-Kitchens. We apply it on frames of HowTo100M sampled at 1 FPS, and then consider all detections exceeding a certain probability threshold as candidates for pseudo ground-truth data. We also check whether the narration associated with the given video frame includes a token corresponding to one of the 295 coarse object categories. If it does, we accept that particular detection as pseudo ground-truth annotation, otherwise we discard it.

This procedure allows us to automatically annotate about $559K$ HowTo100M video frames with a total of about $1.1M$ bounding boxes spanning $154$ object categories. The remaining $141$ EPIC-Kitchens classes are discarded due to yielding too few detections in HowTo100M. As in the original HowTo100M dataset, each of these frames has an associated transcribed narration. Throughout the rest of the paper, we refer to this subset of HowTo100M frames as HowTo100M_BB.

### 3.2 Learning Contextualized Object Embeddings

We now present COBE, our approach for learning contextualized object embeddings from instructional videos of HowTo100M. COBE is comprised of three high-level components: 1) a pretrained contextualized language model which maps the automatically-transcribed narration associated with a given frame to a semantic feature space, 2) a visual detection model (based on a Faster R-CNN architecture) which predicts a contextualized object embedding for every detected object instance, and 3) a contrastive loss function which forces the predicted object embedding to be similar to the contextualized word embedding. Below, we describe each of these components in more detail.

**Contextual Language Model.** Contextualized word embedding models such as BERT [15, 47, 48, 49, 50, 51] have enabled remarkable progress in many language problems. Due to their particular

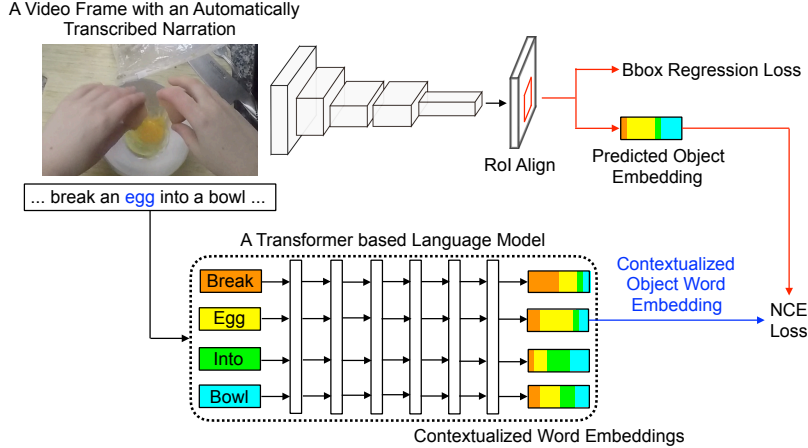

A Video Frame with an Automatically Transcribed Narration

... break an egg into a bowl ...

Bbox Regression Loss

RoI Align

Predicted Object Embedding

A Transformer based Language Model

Break
Egg
Into
Bowl

Contextualized Word Embeddings

Contextualized Object Word Embedding

NCE Loss

Figure 2: An illustration of the procedure used to train COBE. Given an instructional video frame with an automatically-transcribed narration, we process the textual narration with a contextualized language model and feed the video frame to a visual detection model corresponding to a modified Faster R-CNN. We use a contrastive loss to guide the object embedding predicted by the visual model to be similar to the contextualized word embedding obtained from text.

design, such models learn an effective token representation that incorporates context from every other word in the sentence. Conceptually, this is similar to what we are trying to do, except that we want to incorporate relevant visual context for each object instance. Since every video frame in our dataset is accompanied by an automatically-transcribed speech, we want to exploit contextual language models for learning contextualized object embeddings.

Formally, let us consider a video frame $I$ and its accompanying instructional narration in the form of a sequence $x = (x_1, \ldots, x_n)$ where $x_j$ represents a distinct word in the narration and $n$ is the length of the narration segment considered. We feed the textual narration $x$ into a pretrained language model $g$, which outputs a matrix $g(x_1, \ldots, x_n) \in \mathbb{R}^{n \times d}$ where each row corresponds to a contextualized word representation for each word token. For brevity, we denote $g_j$ as the $j$-th row of this matrix. Note that due to the use of the self-attention blocks [52] in these language models, each vector $g_j$ provides a representation of word $x_j$ contextualized by the entire sentence $(x_1, \ldots, x_n)$, thus providing a rich semantic representation of each word *and* the contextual narration where it appears.

In order to train our visual detection model, we use only narration words $x_j$ that match the categorical label of one of the EPIC-Kitchen object classes in the frame. The contextualized word embeddings $g_j$ corresponding to these words are used to supervise our model. We note that because of the design of our augmentation procedure (see steps outlined in 3.1), every frame in HowTo100M_BB contains at least one object instance with its category word represented in the associated narration, thus yielding at least one training vector $g_j$.

**Visual Detection Model.** Our visual detection model is based on the popular Faster R-CNN detector [2] implemented using a ResNeXt-101 [53] backbone with a feature pyramid network (FPN) [54]. The main difference between our detector and the original Faster R-CNN model is that we replace the classification branch [2] with our proposed contextualized object embedding (COBE) branch (see Figure 2). We note that the original classification branch in Faster R-CNN is used to predict a classification score $s_i \in \mathbb{R}^C$ for each Region of Interest (RoI) where $C$ is the number of coarse object categories. In contrast, our proposed COBE branch is trained to predict a contextualized word embedding $f_i \in \mathbb{R}^d$ matching the embedding computed from the corresponding narration. Our COBE branch is advantageous because it allows us to capture not only the coarse categorical label of the object but also the contextual language information contained in the narration, e.g., the object size, color, shape as well as co-occurring objects and actions.

**Loss Function.** We use the noise contrastive estimation criterion [55, 56] to train our visual detection model. Consider an object embedding $f_i \in \mathbb{R}^d$ predicted by our visual model for the $i^{th}$ foreground RoI. Let $g_i^+$ be the *matching* contextual word embedding computed from the narration, i.e., such that the word associated to $g_i^+$ matches the class label of the $i$-th RoI. Now let us assume that we also sample a set of $m$ negative word embeddings that are *not* associated with the category label of object

embedding $f_i$. We pack these negative embeddings into a matrix $H_i \in \mathbb{R}^{m \times d}$. A contrastive loss is then used to measure whether $f_i$ is similar to its positive word embedding $g_i^+$, and dissimilar to the negative word embeddings $H_{ik}$. With similarity measured as a dot product, we can express our NCE loss for the foreground and background RoIs as:

$$\mathcal{L}_{fg} = \sum_{i \sim \mathcal{FG}} -\log \frac{e^{(f_i \cdot g_i^+)}}{e^{(f_i \cdot g_i^+)} + \sum_{k=1}^{m} e^{(f_i \cdot H_{ik})}} \quad \mathcal{L}_{bg} = \sum_{i \sim \mathcal{BG}} -\log \frac{1}{1 + \sum_{k=1}^{m} e^{(f_i \cdot H_{ik})}} \quad (1)$$

where in the left equation the outer summation is over all foreground RoIs (denoted as $\mathcal{FG}$), and in the right equation is over the entire set of background RoIs (denoted as $\mathcal{BG}$). We note that using different losses for the background and foreground RoIs is necessary because we do not want to associate background RoIs to any of the contextual narration embeddings. The final NCE loss function is expressed as: $\mathcal{L} = (\mathcal{L}_{fg} + \mathcal{L}_{bg})/B$, where $B$ is the total number of RoIs. We refer the reader to our supplementary material for further implementation details.

## 4 Experimental Results

**Evaluation Task.** We want to evaluate the effectiveness of our detection model according to two aspects: 1) its accuracy in localizing objects in video frames, and 2) its ability to represent object contextual information. To do this, we propose a new task, called contextualized object detection. This task requires predicting a $(noun, context)$ tuple, and a bounding box associated with $noun$. Here, $noun$ represents the coarse category of an object, whereas $context$ is any other word providing contextual information about that object. A special instance of this task is the detection of co-occurring objects, which requires predicting a $(noun, noun)$ tuple, and a bounding box associated with the first $noun$. For example, predicting $(pan, mushrooms)$ means that a model needs to spatially localize a pan with mushrooms near it (or possibly in it). Another instance of a contextualized object detection task is the object-action detection task, which requires predicting a $(noun, verb)$ tuple, and a bounding box associated with the $noun$. In this case, the predicted noun indicates a coarse object category, while the verb denotes an action that has been applied on that object. For instance, predicting the tuple $(onion, peel)$ means that a model should spatially localize an onion being peeled. In general, $context$ can take any of the following forms: $noun, verb, adjective$ or $adverb$.

**Using COBE to Predict Discrete Labels.** To evaluate our model on the contextualized object detection task, we need to be able to predict probabilities over the discrete space of $(noun, context)$ tuples defined for a given dataset. Given a discrete set of tuples, we first feed all such tuples through the same contextual word embedding model used to train our detector. This allows us to generate a matrix $Z \in \mathbb{R}^{T \times d}$, which contains contextualized word embeddings for all of the $T$ possible tuples. Afterwards, to make a prediction in the discrete tuple space, we feed a given video frame through our visual detection model, and obtain a continuous contextualized object embedding $f_i \in \mathbb{R}^d$ for each object instance $i$. To compute probabilities $p_i \in \mathbb{R}^T$ over the discrete tuple space, we perform a matrix-vector product and apply a softmax operator: $p_i = softmax(Zf_i)$.

**Evaluation Datasets.** Our evaluation dataset needs to satisfy three key requirements: 1) it should involve a variety of human-object interactions, 2) it should be annotated with bounding box labels, and 3) it should contain textual descriptions of the objects beyond mere object categories. The dataset that best fits all of these requirements is the EPIC-Kitchens dataset [16] (denoted from now as EK for brevity). It contains 1) $(noun, verb)$ action tuple labels, 2) bounding box annotations of *active* objects, and 3) a list of $nouns$ associated with an action (e.g., ['pan', 'mushrooms'] for "put mushrooms in the pan"). We construct the ground truth by finding frames where the object category of a bounding box matches the $noun$ of either 1) an action tuple or 2) one of the $nouns$ from a $noun$ list. Furthermore, to expand our evaluation beyond EK, we manually label a collection of frames from HowTo100M and YouCook2_BB [17] with a set of $(noun, context)$ tuple labels. This leads to the following evaluation sets: 1) $9K$ frames with 171 unique $(noun, context)$ tuples in HowTo100M_BB_test, 2) $25.4K$ frames with 178 unique $(noun, context)$ tuples in EK, and 3) $2K$ frames with 166 unique $(noun, context)$ tuples in YouCook2_BB.

**Evaluation Metric.** The performance is evaluated using the widely adopted COCO [10] metric of mean average precision, which is computed using an Intersection over Union (IoU) threshold of $0.5$. In the subsections below, we present our quantitative results.

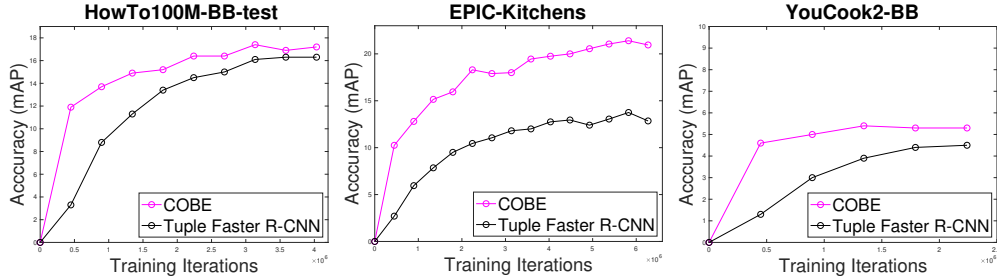

Figure 3: Contextualized object detection results on the evaluation sets of HowTo100M_BB_test, EPIC-Kitchens, and YouCook2_BB. We compare COBE with the Tuple Faster R-CNN baseline, which is adapted to produce detections of the form $(noun, context)$ as the task requires. Both methods use the same network architecture (except for the contextualized object embedding branch), and are trained on the same HowTo100M_BB dataset. Whereas COBE uses contextual word embeddings as its supervisory signal, Tuple Faster R-CNN uses the discrete tuple labels. We study the mAP performance of each method as a function of training iterations. Based on these results, we observe that COBE outperforms Tuple Faster R-CNN on all three datasets.

## 4.1 Comparing COBE to a Discrete Detector

To the best of our knowledge, there are no prior published methods that we can directly include in our comparisons for the contextualized object detection task. We note that our task differs from the action classification task in the original EK benchmark because the latter 1) *does not* require spatial localization of object instances, and 2) it focuses exclusively on $(noun, verb)$ classification.

Thus, as our primary baseline, we consider a method that we name "Tuple Faster R-CNN." Given a video frame as its input, it produces detections of the form $(noun, context)$ just as our task requires. We train this model on the same HowTo100M_BB dataset as our method but instead of using contextual word embeddings as its supervisory signal, it uses the discrete tuple labels. We also note that Tuple R-CNN uses the same network architecture as our COBE, except for the contextualized object embedding branch, which is instead replaced by the standard classification branch.

The performance of both methods is evaluated on the evaluation sets of 1) HowTo100M_BB_test, 2) EK, and 3) YouCook2_BB. We present these results in Figure 3, where we plot the detection accuracy as a function of training iterations. Our results suggest that compared to the Tuple Faster R-CNN, COBE is more effective, and it requires fewer training iterations to achieve strong performance. Furthermore, the fact that COBE outperforms Tuple Faster R-CNN on all three datasets highlights the wide applicability of our learned representation.

## 4.2 Comparison to Models Trained on Large-Scale Manually-Labeled Datasets

In this section, we compare our strategy of training on narrated frames with traditional training on large-scale manually-labeled datasets. Specifically, we include baselines that *combine* a standard Faster R-CNN detector (trained on HowTo100M_BB) with $noun$ or $verb$ classifiers pretrained on large-scale labeled data. To implement highly competitive baselines, we use several state-of-the-art models trained on large-scale manually-labeled datasets: a 2D ResNet-101 trained on ImageNet [27], a 3D ResNet-101 trained on the Kinetics dataset [57], and an R(2+1)D ResNet-152 trained on the Instagram-65M (IG65M) dataset [58] as our image-level or video-level classifiers. We adapt these baselines to the task of contextualized object detection on the EK dataset by generating all possible tuple combinations between the frame-level detections of Faster R-CNN, and the top-5 predictions of the $noun/verb$ classifier. For example, if Faster R-CNN detects $tomato$ and $bowl$ in a given video frame, while the pretrained classifier predicts $(knife, cucumber, cuttingboard, spoon, fork)$ as its top-5 predictions, this would result in 10 distinct tuple predictions: $(tomato, knife), (tomato, cucumber)$, ... , $(bowl, spoon), (bowl, fork)$.

We stress that the Faster R-CNN detector used in all of the baselines is trained on the same HowTo100M_BB dataset as COBE, and it also uses the same network architecture as COBE (except for the contextualized object embedding branch). It should also be noted that we do not finetune or pretrain on EK any of the evaluated models (including ours). Having said this, we do adopt a detector

Table 1: Results on the detection of $(noun, noun)$, and $(noun, verb)$ tuples. The evaluation is performed on different subsets of the EPIC-Kitchens dataset using mAP with an IoU threshold of $0.5$. Our COBE outperforms by large margins several competitive baselines in both settings.

| Method | Training Data | Eval. Set | # of Tuples | # of Frames | Noun-Noun | Noun-Verb |
|---|---|---|---|---|---|---|
| Faster R-CNN + S3D [59] | HowTo100M_BB, HowTo100M | EK_H | 178 | 25.4K | 9.0 | 14.1 |
| Tuple Faster R-CNN | HowTo100M_BB | EK_H | 178 | 25.4K | 11.5 | 14.0 |
| **COBE** | HowTo100M_BB | EK_H | 178 | 25.4K | **16.9** | **24.7** |
| Faster R-CNN + 2D ResNet-101 [27] | HowTo100M_BB, ImageNet | EK_I | 23 | 3K | 13.2 | - |
| **COBE** | HowTo100M_BB | EK_I | 23 | 3K | **26.0** | - |
| Faster R-CNN + R(2+1)D ResNet-152 [58] | HowTo100M_BB, IG65M | EK_K | 43 | 8.6K | 8.4 | 15.0 |
| Faster R-CNN + 3D ResNet-101 [57] | HowTo100M_BB, Kinetics | EK_K | 43 | 8.6K | 7.7 | 19.4 |
| **COBE** | HowTo100M_BB | EK_K | 43 | 8.6K | **32.4** | **30.1** |

trained on EK to pseudo-label the frames of HowTo100M (see Section 3.1), which might ease the transfer to EK. However, this applies to all of our baselines, not just COBE.

We also note that it may be argued that while COBE has been trained on an open-ended continuous distribution, the baselines (except for Tuple Faster R-CNN) have the disadvantage of being tested on some EK tuples that do not exist in the discrete label space of their training set. For example, the Kinetics classifier was trained on $400$ action categories, many of which are not nouns. Thus, to make the comparison with these baselines more fair, we use the following evaluation protocol. For each of our introduced baselines, we construct a subset of EK, where the ground truth $(noun, noun)$ or $(noun, verb)$ tuples include only the categories that the baseline was trained on. This leads to three different evaluation subsets, which we refer to as EK_I (for the ImageNet baseline), EK_K (for the Kinetics and IG65M baselines, which share the same Kinetics label space), and EK_H (including all tuples that appear in both EK and HowTo100M_BB).

We present our quantitative results in Table 1. Based on these results, we observe that COBE outperforms by a large margin all of the baselines. This suggests that our proposed framework of learning from narrated instructional videos is beneficial compared to the popular approach of training on large manually-labeled datasets such as ImageNet, Kinetics or IG65M. We also include in this comparison the recent S3D model [59] trained via joint video-text embedding on the HowTo100M dataset. This is a relevant baseline since, similarly to our approach, it leverages the "free" supervision of narration to train the visual model. However, unlike our approach, which trains the visual model as a contextualized object detector, it learns a holistic video embedding unable to localize objects. As shown in Table 1, this S3D baseline [59] combined with the Faster R-CNN detector achieves much lower accuracy than COBE. This indicates that holistic video models such as [59] cannot be easily adapted to this task, and that a specialized contextualized object detector such as COBE might be better suited for solving this problem.

### 4.3 Zero-Shot and Few-Shot Learning

We also consider the previous task in the scenarios of zero-shot and few-shot learning. We use 30 EK $(noun, noun)$ categories and 26 EK $(noun, verb)$ categories that have not been seen during training. We refer to this subset of EPIC-Kitchens as EK_Z to indicate that it is used for *zero*-shot learning experiments. Furthermore, in order to investigate the *few*-shot learning capabilities of our model, we randomly select 5 examples for each unseen tuple category (from the data that does not belong to the EK_Z evaluation set), and then finetune our model on these examples.

Most of our baselines from the previous subsection are unfortunately not applicable in this setting because here we only consider tuple categories that have not been previously seen by any of the classifiers. Thus, in this subsection we can only compare our model in the few-shot setting with a Tuple Faster R-CNN that was trained on HowTo100M_BB and finetuned on the few examples of EK (this baseline is inapplicable in the zero-shot setting). From Table 2, we observe that in the zero-shot setting, COBE produces reasonable results even though it has never been trained on these specific tuple categories. In the few-shot setting, COBE outperforms Tuple Faster R-CNN, demonstrating a superior ability to learn new categories from few examples.

### 4.4 Object Detection Results

We also conduct object detection experiments on the $124K$ frames of EK (180 object categories) by comparing COBE to a Faster R-CNN traditionally trained for object detection. Both methods share

Table 2: We evaluate few-shot and zero-shot detection performance on a subset of EPIC-Kitchens (EK) containing 30 $(noun, noun)$, and 26 $(noun, verb)$ tuple-categories that were not included in the training set. For few-shot learning experiments, we finetune both methods on the EK data with 5 samples per tuple-category. Despite the challenging setting, our method performs well on both tasks.

| Method | Training Data | Eval. Set | # of Tuples | # of Frames | Noun-Noun | | Noun-Verb | |
|---|---|---|---|---|---|---|---|---|
| | | | | | Zero-Shot | Few-Shot | Zero-Shot | Few-Shot |
| Tuple Faster R-CNN | HowTo100M_BB | EK_Z | 56 | 15.6K | - | 0.3 | - | 0.1 |
| **COBE** | HowTo100M_BB | EK_Z | 56 | 15.6K | **10.3** | **25.5** | **8.6** | **29.5** |

the same architecture (except for the contextualized object branch). Also, note that both methods are trained on HowTo100M_BB, and not on EK. We evaluate the performance of each method using mAP with an IoU threshold of 0.5. We report that COBE achieves **15.4** mAP whereas Faster R-CNN yields **14.0** mAP in the detection accuracy. We also note that pre-training COBE on HowTo100M_BB and then finetuning it on EK produces a mAP of **22.6**, whereas COBE only trained on EK yields a mAP of **12.7**. This highlights the benefit of pretraining on HowTo100M_BB.

## 4.5 Qualitative Results

**Object-To-Text Retrieval.** Since COBE produces outputs in the same space as the contextualized word embedding, we can use it for various object-to-text retrieval applications. For example, given a predicted object embedding, we can compute its similarity to a set of contextualized word embeddings of the form $(object, context)$ where $object$ represents a coarse object-level category from EK and $context$ is any other word providing contextual details. In Figure 4, we visualize some of our results, where the green box depicts a detected object, which is used as a query for our text-retrieval task. Our results indicate that the text retrieved by our model effectively captures diverse contextual details for each of the detected objects. Furthermore, we would like to point out that these results are obtained on the EK dataset, without training COBE on it. Despite the semantic and appearance difference between HowTo100M and EK, COBE successfully generalizes to this setting.

**Text-To-Object Retrieval.** We can also reverse the previous object-to-text retrieval task, and instead use text queries to retrieve object instances. In Figure 5, we visualize the top retrievals for several $(object, context)$ queries. Each column in the figure depicts the $object$ part of the tuple, while different rows illustrate different $context$ words in the the tuple. Our results suggest that COBE captures a rich variety of fine-grained contextual cues, including the states of the objects, their functional properties, color, and shape, as well as the actions applied to them.

**Visual Object Analogies.** Prior work on language representation learning [60] has shown that it is possible to perform semantic analogies among different words by adding/subtracting their vector embeddings. Since our model is trained to predict a contextualized word embedding for each object instance, we consider the same arithmetic operations with COBE in order to leverage the compositional properties of language models but in the visual domain. In Figure 6, we visualize a few examples of visual object analogies. We build COBE queries by adding the difference between two COBE vectors to a third one, and then perform retrieval of objects in EK. Our examples demonstrate

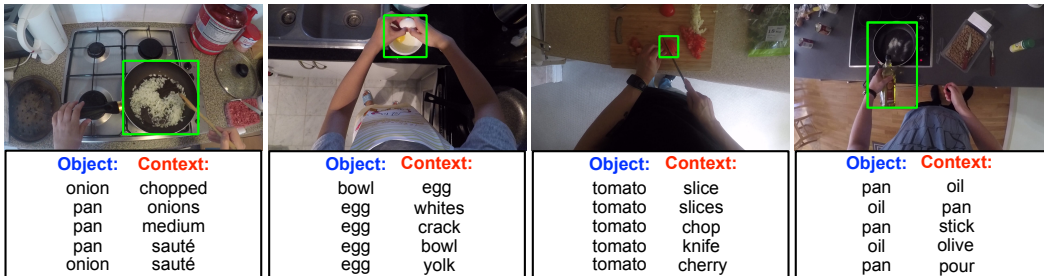

Figure 4: Examples of object-to-text retrieval on EPIC-Kitchens. Given a visual object query (denoted with a green bounding box), we retrieve the $(object, context)$ pairs that are most similar to our COBE visual features in the space of the contextualized language model. The retrieved text examples show that our visual representation captures rich and diverse contextual details around the object.

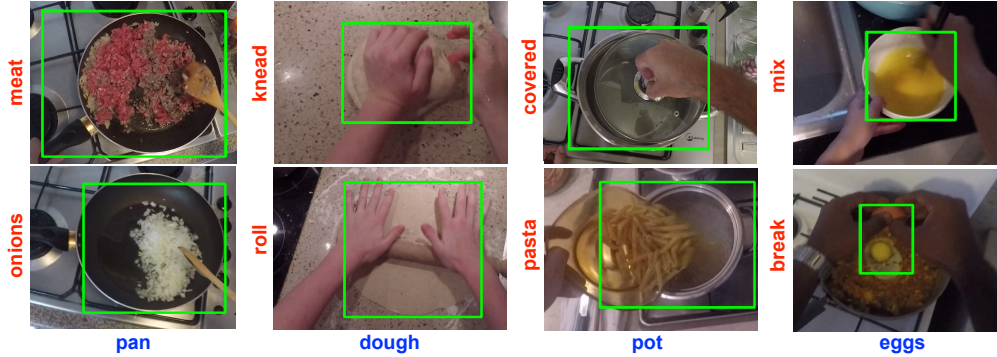

Figure 5: Qualitative illustrations of text-to-object retrieval (cropped for better visibility). Given a textual query of the form (*object*, *context*), the method retrieves the most similar COBE object instances in the space defined by the contextualized language. Note that each column shows different contexts for a fixed object.

that we can meaningfully combine visual concepts via subtraction and addition in the learned contextual object embedding space. For instance, the example in the first row of Figure 6 shows that adding the difference between "cutting dough" and "dough" to a "fish" image in COBE space yields the retrieval of "cutting fish." Similarly, the example in the second row shows that adding the difference between "pouring milk into a glass" and "pouring water into a glass" to a "a coffee cup" in COBE space results in a retrieved "pouring milk into a coffee cup" object. These examples suggest that our system retains some compositional properties of the language model that was used to train it.

### 4.6 Ablation Studies

In our supplementary material we present ablation experiments studying how 1) different contextualized language models and 2) the negative-to-positive sample ratio affect performance.

## 5 Discussion

In this work, we introduced COBE, a new framework for learning contextualized object embeddings. Unlike prior work in this area, our approach does not rely on manual labeling but instead leverages automatically-transcribed narrations from instructional videos. Our experiments demonstrate that COBE learns to capture a rich variety of contextual object cues, and that it is highly effective in zero-shot and few-shot learning scenarios.

COBE leverages a pretrained object detector to generate pseudo annotations on videos. While this removes the need for manually labeling frames, it limits the approach to predict contexts for only the predefined set of object classes recognized by the detector. Extending the method to learn contextualized object detection from instructional videos without the use of pretrained models is challenging, particularly because captions are noisy. However, solving this problem would allow us to take a big step towards a self-supervised learnable detection framework. We intend to tackle this problem in our future work.

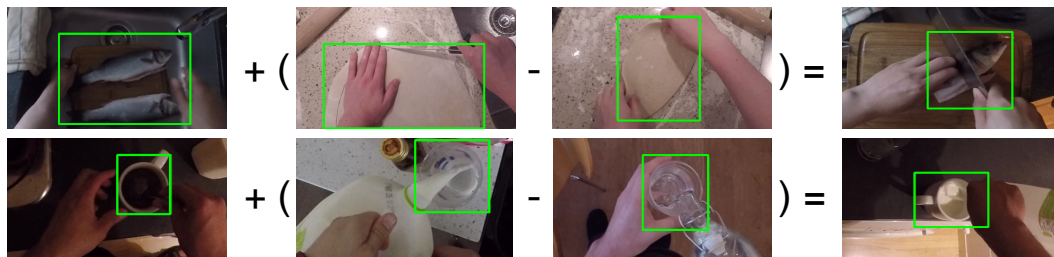

Figure 6: Visual object analogies: just like prior work on language representation learning [60], we show that we can leverage our learned contextualized object embeddings to combine different visual concepts via simple vector arithmetics. The last column shows object instances retrieved based on visual queries defined by arithmetics in the COBE space.

## Broader Impact

This work considers the learning of object embeddings from narrated video. In terms of positive impact, our system is particularly relevant for language-based applications on unlabeled video (e.g., text-based retrieval) and may facilitate a tighter integration of vision and NLP methods in the future.

As for the potential negative effects, we note that COBE learns to capture various aspects of human actions. Thus, because we are using videos from an uncurated Web dataset, it is possible that COBE might learn contextualized representations that are subjective towards particular characteristics. Furthermore, we note that any algorithm that is relevant to action recognition–and to retrieving videos based on arbitrary language queries–may potentially be used for surveillance purposes.

## Funding Transparency Statement

We did not receive any third-party funding in direct support of this work.

## Acknowledgements

We would like to thank Fabio Petroni, and Rohit Girdhar for helpful discussions. Additionally, we thank Bruno Korbar for help with the experiments.

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
