[Supplementary Material]

# COBE: Contextualized Object Embeddings from Narrated Instructional Video

## Supplementary Materials

**Gedas Bertasius**[1], **Lorenzo Torresani**[1,2]
[1]Facebook AI, [2]Dartmouth College

Our supplementary materials consist of:

1. Implementation Details.
2. Ablation Sudies.

## 1 Implementation Details

We train our model for 10 epochs with an initial learning rate of 0.001, a linear warmup of 500 steps and a momentum of 0.9. The hyperparameters of RPN and FPN are the same as in [1]. We use a multi-scale training approach implemented by resizing the shorter side of the frame randomly between 400 and 800 pixels. Our model is trained in a distributed setting using 64 GPUs, each GPU holding a single frame. We initialize our model with a Faster R-CNN pretrained on COCO for object detection. During training, we use $m = 2048$ randomly-sampled negative contextualized embeddings. As our contextual language model, we use a pretrained Conditional Language Transformer Model (CTRL) [2], which we found to be slightly more effective than the other state-of-the-art language models [3, 4, 5, 6, 7]. We freeze the weights of this language model, remove its last layer and add to it a shallow 5-layer MLP which we train jointly with our visual detection model. The contextualized embedding vectors have dimensionality 1280. During inference, we run the bounding box prediction branch on 1000 proposals, apply non-maximum suppression, and use boxes with a score higher than 0.001 as our final output.

## 2 Ablation Studies

**Contextual Language Model.** To investigate the effectiveness of different language models, we evaluate COBE on the manually annotated test split of HowTo100M_BB using the following state-of-the-art language models: RoBERTa [5], T5 [8], ELECTRA [9], Transformer XL [10], BERT [3], XLNet [6], ALBERT [4], Word2Vec [11], XLM [7], and CTRL [2]. As before the performance of each model variant is evaluated according to the standard mAP detection metric. We present these results in Table 1. These results indicate that the choice of contextual language model in our system can be quite important as the results in Table 1 range from 9.4 mAP to 18.4 mAP, which is a substantial gap. We select the Conditional Transformer Language Model (CTRL) [2] as our choice as it exhibits the best performance on our validation set.

**Negatives per Positive.** Prior work [12, 13] has demonstrated that using a large number of negative samples per positive sample is important for approaches that leverage NCE loss [14]. Here, we validate this finding in our setting and present these results in the last three rows of Table 1. Specifically, we experiment with the negative to positive sample ratio of 128, 512 and 2048. We observe that using a large number of negative samples per one positive sample yields considerably better performance. We

Table 1: Here, we study the effectiveness of different language models used to train our visual detection model. The ablation studies are conducted on the test set of HowTo100M_BB dataset. We evaluate the performance of each baseline using the same mean average precision metric as before. Based on these results, we note that the Conditional Transformer Language Model (CTRL) [2] achieves the best accuracy so we adopt it as our contextual language model. Furthermore, we also investigate how the negative-to-positive sample ratio during training affects our model's performance. As expected, a larger number of negative per single positive sample leads to better results.

| Language Model | Training Data | Negatives per Positive | Eval. Set | mAP |
|---|---|---|---|---|
| RoBERTa [5] | HowTo100M_BB | 2048 | HowTo100M_BB_test | 9.4 |
| T5 [8] | HowTo100M_BB | 2048 | HowTo100M_BB_test | 14.1 |
| ELECTRA [9] | HowTo100M_BB | 2048 | HowTo100M_BB_test | 14.5 |
| Transformer XL [10] | HowTo100M_BB | 2048 | HowTo100M_BB_test | 15.3 |
| BERT [3] | HowTo100M_BB | 2048 | HowTo100M_BB_test | 16.2 |
| XLNet [6] | HowTo100M_BB | 2048 | HowTo100M_BB_test | 16.3 |
| Word2Vec [11] | HowTo100M_BB | 2048 | HowTo100M_BB_test | 16.7 |
| ALBERT [4] | HowTo100M_BB | 2048 | HowTo100M_BB_test | 16.9 |
| XLM [7] | HowTo100M_BB | 2048 | HowTo100M_BB_test | 18.0 |
| CTRL [2] | HowTo100M_BB | 128 | HowTo100M_BB_test | 13.6 |
| CTRL [2] | HowTo100M_BB | 512 | HowTo100M_BB_test | 15.5 |
| CTRL [2] | HowTo100M_BB | 2048 | HowTo100M_BB_test | **18.4** |

note that we have not experimented with even larger negative samples as it slows down the training, but we will attempt to do so in our future work.