[Reviews · NeurIPS 2020]

Review 1

Summary and Contributions: This paper deals with the problem of detection beyond nouns and verbs in video: for example, one might want to detect "onions being cut," "eggs in a bowl," or "tomato sauce". Rather than attempt to label all such phrases, this paper uses the natural language provided in the howto100m dataset as supervision. The method first labels this dataset automatically with bounding boxes using a detector pretrained on epic kitchens, and then contextual information for those objects (e.g. actions, adjectives) are inferred from speech in an unsupervised manner as extra labels for the bounding boxes. The system is evaluated by transferring the detectors back to epic kitchens and predicting tuples (generally an object along with some modifier to that object, where the modifier is inferred from the narration included in epic kitchens). Performance is substantially better than algorithms trained to directly predict the modifier, or trained directly to predict the tuples, especially in few-shot scenarios.

Strengths: This is an important problem: existing object detectors are clearly inadequate for understanding complex videos where people take actions to modify attributes of objects. The approach taken in this paper is relatively straightforward and outperforms the baselines. Furthermore, this performance is under a situation of rather large domain shift: the videos used for training (howto100m) have a very different perspective than those used for testing (epic kitchens)

Weaknesses: The overall methodology isn't terribly different from other work on connecting language and video. While this algorithm is specifically designed for detectors, Miech et al 2019 used unsupervised NCE losses (much like the ones in this paper) in order to understand the natural language descriptions associated with videos; the algorithm presented here seems like the most straightforward extension of this idea to bounding boxes. Little attention is given to demonstrating that the use of bounding boxes fundamentally changes the problem. == Update == The rebuttal addresses the following point regarding the accuracy of the evaluation. I had misunderstood the annotations that are available with epic kitchens, and therefore I am changing my review. I would encourage the authors to clarify the writing regarding what's available with epic kitchens. Furthermore, the use of bounding boxes introduces problems with the evaluation. Epic kitchens wasn't designed for this kind of of description of bounding boxes, and therefore does not have phrase or modifier labels. Therefore, the 'ground truth' for the boxes is inferred from the narration, which means it might not be correct. For example, it seems quite possible that a scene could contain two spoons, one being used for stirring and one sitting on the counter; then the 'ground truth' for the spoon on the counter might be labeled as stirring just because the narration mentioned stirring. Considering that the whole justification for this benchmark being separate from classification benchmarks, this seems like a very important detail; however, very little of the text is devoted to ensuring that the benchmark is actually measuring what's intended. In many ways, I would argue that a successful benchmark could be a larger contribution than the model itself, if the paper did a good job convincing readers that the benchmark is really measuring attributes of objects despite no explicit labels for this. The paper also makes the statement that this algorithm wasn't trained on the epic kitchens dataset. However, the training algorithm did use the epic kitchens dataset to pretrain the detectors; therefore, it may only be confidently detecting object instances that look like the ones in epic kitchens. This therefore raises questions about the generalization ability of the overall algorithm. In my opinion, the claims about not being trained on epic kitchens should be qualified more carefully than they are.

Correctness: Reasonably correct, although it's not entirely clear that the benchmarks are measuring what is claimed.

Clarity: For the most part. I overall found the section on the evaluation benchmark (section 4.0) quite difficult to follow, and would encourage the authors to revise and give more detail.

Relation to Prior Work: Yes

Reproducibility: Yes

Additional Feedback:


Review 2

Summary and Contributions: The paper presented an interesting approach for learning contextualized visual embeddings for objects using instructional videos and their transcripts. These embeddings encode the visual context of object regions by matching the detected object features in a video to the corresponding word embeddings from the video’s transcript. To evaluate their method, the authors proposed two context prediction tasks using human object interaction videos. The results are quite promising. The proposed method provides a simple method of visual representation learning that is beneficial to understand objects in their visual contexts and might be useful for vision and language tasks.

Strengths: * The idea of contextualized object representation is interesting and well motivated. * The idea of learning the contextualized object representation using instructional videos has not been explored before. * Some of the results are quite impressive.

Weaknesses: My main concern of the paper is the experimental design. While the experiments are extensive, I am not convinced by the current design and setup. * I thought that a first experiment is to evaluate the object detection performance (“accuracy in localizing objects” as argued in L185 page 5). The expectation is that by modeling contextual cues, the representation might give better performance for detecting objects. And the same method described in L202-220 can be used for object detection. I am not sure why this experiment was not included in the first place. I understand that the proposed method started from a detector pre-trained on EPIC-Kitchens dataset, which could create issues in evaluation. But what if you combined the learned representation and the initial detector? * The current experiment setup is rather confusing. And I have a number of questions. ** I don’t see a major difference between the two proposed tasks. It looks like both tasks are about detecting co-occurring contextual tokens. Whether that token is a noun or a verb does not seem to play a major role. The only difference seems to be the inclusion of baseline methods. ** The softmax in L210 seems to suggest only a single tuple is considered for a given object region. What if you have multiple valid tuples, especially for task 1, where multiple (>=2) nouns can appear in the same transcript? ** The argument in L213 that the proposed method can’t be used for action recognition in EPIC-Kitchens dataset is confusing. I thought the method can provide action recognition results by e.g., selecting the tuple associated with the most confident object region. It is way more convincing to report this result and compare it to existing ones on EPIC-Kitchens dataset. * It seems like an important baseline is missing. As the proposed method started from a EPIC-Kitchens pre-trained detector, an alternative solution is to simply match detection output to the tokens. What if you run the detector on the frames and match the embedding of the detected nouns to the embeddings of the target context token? Other comments: * While the idea is interesting, the technical components (BERT and contrastive learning) are not terribly new, weakening the technical contribution a bit. * I personally found the description in L61-62 misleading. The proposed method does require a detector pre-trained on EPIC-Kitchens before the training on HowTo100M dataset and the application to EPIC-Kitchens dataset.

Correctness: The technical components are sound yet the experiment design has some issues (see my comments in weakness).

Clarity: The paper is reasonably well written.

Relation to Prior Work: Related work is sufficiently discussed.

Reproducibility: Yes

Additional Feedback: I like the idea of the paper and I am in favor of accepting the paper. However, the experiment does not seem very convincing to me. I’d like to see the authors’ response to my questions on the experimental design. ** Post Rebuttal Update: The rebuttal has addressed most of my previous concerns on the experiments. The new object detection results was a bit disappointing, but otherwise the new results look quite convincing. To this end, I have raised my rating to accept. I would urge the authors to revise the experiments and merge the new results in the camera ready.


Review 3

Summary and Contributions: This work proposes to learn visual object embeddings that are contextualized by text associated with each image. The idea is to leverage narrated instructional videos where the instructions are usually well aligned with the video frame contents. They train an object detector to produce an embedding per object bounding box that is close to the category name embedding of that object contextualized from the associated text of the video frame. The language-based embedding comes from a pre-trained language model (Transformer). The model is evaluated on the EPIC-Kitchens (EK) dataset for detecting noun pairs (e.g. [tomato, pan]) or noun-verb pairs (e.g. [tomato, cut]) in standard setting, few-shot learning and zero-shot learning. The model demonstrate better performance than some proposed baselines. ====== Update ====== The rebuttal addressed my questions adequately. I find the submission to be a good application paper with interesting problem domain and results, but with limited technical novelty. I encourage the authors to incorporate the baselines, the discussion of limitations and the comparisons from the rebuttal in the submission to present a stronger case for the proposed model. I recommend 6: Marginally above the acceptance threshold.

Strengths: I find the proposed idea quite simple and straight forward. It demonstrates an interesting application of the recent advances in language models in connection to vision. The model shows reasonable performance on EK when compared to the considered simple baselines. Since the model is based on distributed language representations, it has the ability to generalize to zero-shot and few-shot learning. Overall, the paper is well written and easy to read and understand.

Weaknesses: Novelty: 1- Using word embeddings produced from transformers to guide the learning of the visual embeddings and using videos with aligned textual instructions seems to have been addressed before for an object grounding task [44]. This render the novelty of this work to be quite limited. Furthermore, there is no technical novelty. The model is a direct and simple application of an off-the-shelf transformer (CTRL) and object detector (Fater R-CNN). Baselines: 2- I find the considered baselines quite simplistic. None of the considered baselines leverage a different language representation similar to the proposed model. A baseline that uses the older word representations (like those mentioned in L.91-92 but at the bounding box level) is needed to understand the impact of context that is leveraged by the transformer on embedding learning and generalization. Another baseline to consider is to use simple arithmetics on the word embeddings in the associated text to produce a simplified contextualized representation, for example by using a weighted average like 0.8*w(object) + 0.2*sum(w(rest of words)) where w(.) is the word embedding function. Such a baseline can tell us how much the heavy weight transformer is really contributing for this task. 3- From what I understood from the baselines description, the ResNet classifiers are not trained on HowTo100M. This is quite limiting since the proposed model has the advantage to adapt to the joint distribution of linguistic and visual data. A better alternative would be to fine tune these models based on the word vocabulary of the HowTo100M dataset such that the classifier predicts the words associated with each frame. This could be a more reasonable baseline to compare against. Evaluation: 4- An ablation study on the performance of the model when used to retrieve a single object or verb is missing. I understand that the focus here is on compositional/contextual categories, however, I think it is important to know if there is degradation or improvement in performance (if any) for the non-compositional case. Since the model learns to detect a tomato in different forms and states then it should probably do better in just simply detecting a tomato. What is the performance of the model in this setting compared to the baselines? 5- There is no discussion of model limitations in the submission. While the model is leveraging instructional videos without manual annotations, the pseudo labels are generated from an object detector pre-trained on EK which will make the model restricted by the predefined categories and annotation errors produced by wrong detections. This work doesn't address such issues nor acknowledge the limitations. There is no examples of failure cases produced by the model in the qualitative analysis.

Correctness: In general the claims and methods are correct. There is concerns regarding the considered baselines for evaluation, see Weaknesses for details.

Clarity: Yes.

Relation to Prior Work: The related work section seems adequate.

Reproducibility: Yes

Additional Feedback:


Review 4

Summary and Contributions: This paper proposes a method to be able to localize objects from video frames. The first contribution lies in the fact that instead of predicting a single class for the localized object as standardly done, the model predicts an embedding aligned with natural language description of the object (Contextualized Object Embeddings). This enable to predict richer outputs than single class, for example to be able to predict state/attribute of objects (e.g. a green tomato/ a red tomato) or combination of objects (e.g. a tomato in a bowl). The second contribution is that the authors propose a way to learn this object detector from readily available source of data where language descriptions are obtained from automatic speech recognition. Arguably this method enables to avoid having to manually labelled all possible objects attributes and states while also presenting challenges (since narrations are more noisy than manually provided labels). Finally, the authors propose an evaluation of their method on the Epic Kitchen dataset and demonstrate how the approach can be used to perform zero-shot and low level recognition of novel classes and attributes.

Strengths: - While the idea of aligning images and language description is not novel, to the best of my knowledge, it is the first time I see it applied in the context of object detection which I think is a nice application. - I see that the HowTo100M_BB dataset could be useful for further study, it will be good if the authors confirm that they plan to release the dataset. - The transfer from learning from YouTube videos to EpicKitchens is nice to see as it is was not obvious this would work. - Overall the paper is clearly presented and the method is technically sound.

Weaknesses: * Reliance on bounding box annotations from EpicKitchen: I have a few concerns about the way the HowTo100M_BB was created that I want to share in the context of this work. - 1) Might ease the transfer to EpicKitchen: L230-231 it is said "without any training done on EK". However a detector trained on Epic Kitchen was used to filter the frames from HowTo100M that are used for training. While I agree this is not direct training, that could have helped ease the transfer between the two datasets since we know already that the selected frames are well recognized by a model trained on EK. It would be nice for the authors to comment on that point. - 2) Another limitation is that the method still relies on a pretrained object detector. And in particular can the method really generalize to objects in states that were not annotated in the original bounding box set? (for example if the model does not recognize a `yellow tomato` as being a tomato in the first place, then how can the model recover from that? I understand the model could eventually generalize to `yellow tomato` if the adjective `yellow` was never *pronounced in the captions* together with the word `tomato`, however it seems there is still an assumption that the detector should recognize a `yellow tomato` as being a tomato in the first place. In other words, are you not limited to state of objects that were already existing in the original Epic Kitchen dataset that was used to train the detectors? This sounds to me like an important limitation but I might be missing something here. Do you have example where the model successfully generalize for that use case? In general, it would be great if one could not rely so much on the bounding box annotation but instead discover that implicitly from the data. This bring me to my second question, how exactly do you define L255-256: `have not been seen during training`. Does it mean that the tuples never occurs together in the captions from HowTo100M? In that case, you cannot be sure that the tuple didn't occur visually but was just not described, is that correct? (this relates to my comment above) * As far as I understand the model only uses still frames. Have you considered using a 3D backbone? It seems to me that the temporal cues should also be used to better inform the context of the objects. Another potential gain from using larger temporal context is the misalignment between the captions and what is going on in the videos. Can the author comment on that last point if it was an issue when creating HowTo100M BB (and if yes have you devised any strategies against that misalignment?) * There seems to be context only on the language side: what I felt slightly unclear is why the visual features only comes from the RoI align features? In particular would it make sense to also combine this with a global features describing the whole image in order to be able to better predict the context of the object? (imagine relations where you need to see for example more of the person manipulating the object to really understand what is going on vs having a very tight bounding box). Have you considered a variant of this? (See VilBERT in the Related Work section for example). * Clarification for the training of the FasterRCNN for the baseline: in that case the FasterRCNN is simply trained to predict the class from Epic Kitchen directly, is that correct? Also are you finetuning everything end to end when training COBE ? * Related work: the following work are relevant and should be discussed: - ViLBERT: Pretraining Task-Agnostic Visiolinguistic Representations for Vision-and-Language Tasks: this share some similarities with this approach (jointly using contextual embeddings from image and text). They also rely on object detector and address the point above (context on the image side) by also having a transformer on the vision side. Such method could also be tried here (e.g. by having a transformer on all the detected objects). It is important that the authors clarify the distinction of their work with this. There are other works along this line such as Uniter for example. - End-to-End Learning of Visual Representations from Uncurated Instructional Videos, Miech et al.: CVPR20. I know that CVPR technically happened after the submission but I thought that is a relevant paper anyways (and appeared online before the submission) that could be discussed to better highlight the contribution of the paper. In particular one potential interesting baseline would be to take the model released there (in the same way as you use a model trained on Kinetics) but here the model is able to take any combination of words possible. That could highlight the importance of actually COBE (I think that this would be a stronger baseline than the Tuple Faster RCNN). Again I understand this is not necessarily a fair thing to ask since the paper was technically published after the deadline, but that would make the paper more convincing in my opinion. In addition the loss used here is quite similar to the loss used in the paper as well.

Correctness: Yes.

Clarity: Yes.

Relation to Prior Work: See Weaknesses section.

Reproducibility: Yes

Additional Feedback: SUMMARY: Overall the paper introduces an interesting idea with a good method. The experiment evaluation highlights the benefits of the method, though its hard to tell due to the lack of established baselines for the problem (I suggested one additional baseline and also it would be important to make sure that the training dataset is going to be released as well as a clear evaluation protocol). At the moment I am leaning towards accept but except the authors to address my comments during the rebuttal. ==== POST REBUTTAL ==== I have read the other reviews and the rebuttal. I believe the authors have correctly answered my concerns as well as the ones raised by other reviewers (additional baseline with the model trained on HowTo100M for example as well as some clarifications that will IMO improve the quality of the paper). For these reasons I wish to stand by my acceptance rating.

[Author Response · NeurIPS 2020]

**EK Benchmark May Not Be Correct (R1).** We do not derive the EK ground truth from noisy narrations. The EK dataset contains 1) $(noun, verb)$ action tuple labels, 2) bounding box annotations of *active* objects, and 3) a list of $nouns$ associated with an action (e.g., ['pan', 'mushrooms'] for "put mushrooms in the pan"). We construct the ground truth by finding frames where the object category of a bounding box matches the $noun$ of either 1) an action tuple or 2) one of the $nouns$ from a $noun$ list. We manually verified our annotations and confirmed that they were error-free.

**Claims of Not Training COBE on EK (R1, R2, R4).** We acknowledge that even though we do not directly train COBE on EK, we use a detector trained on EK to pseudo-label the frames of HowTo100M, which might ease the transfer to EK. We will revise our former claims to make this point clear. Also, note that we do *not* start from a detector pre-trained on EK when training on HowTo100M_BB. To expand our evaluation beyond EK, we also manually labeled a test set of $9K$ HowTo100M frames (using a disjoint set from HowTo100M_BB) with 171 unique $(noun, context)$ tuples, where $context$ can be one of: $noun, verb, adjective$ or $adverb$. On this new set, COBE achieves 17.5 mAP, while Tuple Faster R-CNN yields 16.1 mAP. We will release all benchmark annotations upon publication of the paper.

**Additional Baselines (R1, R2, R3, R4).** As requested by R3, and R4, we use S3D trained on HowTo100M [Miech et al., CVPR20] with Faster R-CNN to construct a baseline for our task (as for the other baselines in Table 1 of our draft). While this baseline performs better than other action recognition baselines, its mAP accuracy is 7.9 and 10.6 lower than that achieved by COBE on the EK_H evaluation set for $(noun, noun)$, and $(noun, verb)$, respectively. In response to R1's question, we note that these results indicate that classification systems such as those of Miech et al. cannot be easily adapted to our problem, and that a specialized design such as COBE might be needed for better solving this task. Also, as suggested by R3, we replaced CTRL, with a word2vec embedding, which produced mAP accuracies of 15.8 and 21.9 compared to the 16.9 and 24.7 of COBE w/ CTRL. We also tried the weighted averaging scheme suggested by R3, which yielded 14.9 and 22.1 in mAP. Lastly, per R2's request, we implemented a baseline that "matches object detection outputs to noun tokens," which achieved a 6.7 mAP on the EK_H (compared to the 16.9 mAP of COBE).

**Object Detection Results (R2, R3).** As requested, we ran object detection experiments on $124K$ frames of EK (180 object categories) by comparing COBE to a Faster R-CNN trained on HowTo100M_BB for object detection. Both methods share the same architecture (except for the contextualized object branch). COBE outperforms this baseline by 1.4 mAP. We also note that pre-training COBE on HowTo100M_BB and then finetuning it on EK outperforms Faster R-CNN only trained on EK by 10.1 mAP. This highlights the benefit of pretraining on HowTo100M_BB.

**Limitations of COBE (R3, R4).** R3 and R4 correctly point out that COBE relies on a predefined set of noun-centric object categories. However, because of the supervision from narration, our detector can recognize fine-grained contextual cues beyond the coarse categorical labels. For example, a standard object detector can predict that there is a "tomato" in an image, whereas COBE can predict that there is "a yellow tomato" or "a chopped tomato." R4 rightly points out that unseen tuple categories may still occur visually in the training frames even though they are not mentioned in the text. However, because COBE is supervised by textual captions, it would *not* be trained to predict the categories of those "unmentioned" tuples. Thus, in those cases, COBE is indeed inferring new tuple concepts. We fully agree with R3 and R4 that it would be great to eliminate the reliance on bounding box labels. However, this is challenging, particularly because HowTo100M captions are noisy. We intend to tackle this problem in our future work.

**Related Work (R4).** ViLBERT is better suited for tasks that require multi-modal inputs (e.g. VQA, VCR). Instead, COBE uses captions to supervise a visual detection model during training, but then operates on video frames alone during inference. We thank R4 for highlighting these related models, which we will cite and discuss in our final paper.

**EK Action Recognition Results (R2).** Unlike action recognition models, which are trained on manually annotated action labels, COBE is trained on noisy narrations. Thus, such a comparison would not be properly fair. However, we agree with R2 that such experiments would be useful and we will attempt to include them in our final draft.

**Technical Novelty (R1, R2, R3).** We acknowledge that our work offers limited contributions in terms of model design. Our main technical contribution is our large-scale training framework which leverages noisy narrations for learning object detectors that generalize to novel states. This is in contrast to prior work [43, 44] that focuses on noun-centric object detection, and that relies on smaller datasets with manually annotated text descriptions. As noted by R4, our idea is simple, and effective, and it demonstrates a novel object detection application of modern language models.

**Experimental Questions (R2, R4).** 1) Prior to our evaluation, we verified that our evaluation set does *not* contain frames with multiple tuple categories. However, we agree with R2 that for instances with multiple valid tuples, using a softmax would be suboptimal. To address this issue, we would use a sigmoid instead of a softmax in our formulation in L210. 2) The two tasks are indeed separated to include specialized baselines. 3) We considered a 3D backbone, but it was too costly to train it on HowTo100M_BB. 4) Misalignment between captions and videos was not an issue because we used only the most confident detections to construct HowTo100M_BB (see L118-L120). 5) We appreciate R4's advice of using global features for capturing more context: we will try it. 6) We confirm that: a) Faster R-CNN is trained to predict the classes from EK, and b) COBE is trained end-to-end.

[Meta-Review · NeurIPS 2020]

Paper originally received fairly positive ratings from the reviewers: 6, 6, 6, 7. Reviewers thought that the paper addressed an important problem [R1], the core idea of contextualized object representations was interesting and well motivated [R2], the approach was intuitive/straightforward [R1,R2] and outperformed the baselines [R1,R2]. At the same time reviewers have raised a number of concerns, mainly: (1) lack of methodological novelty [R1,R3,R4], (2) lack of experimental design [R2,R3] and need for additional baselines [R3], and (3) the use of pre-trained object detectors [R1,R4]. Rebuttal was provided by the authors to address these concerns, which included additional baselines requested by reviewers. Reviewers found rebuttal nearly unanimously compelling, with R2 raising the score from 6 to a 7, leading to the final scores for the paper of: 6, 7, 6, 7. AC has read the reviews, rebuttal and the paper itself and largely agrees with the assessment provided by the reviewers. While technical novelty of the approach may indeed be somewhat limited, the core idea is interesting, effective and the application domain challenging and novel; the paper is also well written and the experimental results are robust. Therefore the final decision is Acceptance. NOTE FROM PROGRAM CHAIRS: For the camera-ready version, please expand your broader impact statement to discuss the potential negative impacts of your work. As one reviewer notes, "this algorithm is learning about human actions from an uncurated web video dataset. The potential for learning problematic biases is enormous, considering that web videos tend to depict different races/genders doing different things, and algorithms are well known to reproduce and even amplify these biases. Furthermore, any algorithm that's relevant to action recognition--and to retrieving videos based on arbitrary language queries--has applications in authoritarian surveillance."